# The Effects of Short-Term PM_2.5_ Exposure on Pulmonary Function among Children with Asthma—A Panel Study in Shanghai, China

**DOI:** 10.3390/ijerph191811385

**Published:** 2022-09-09

**Authors:** Ji Zhou, Ruoyi Lei, Jianming Xu, Li Peng, Xiaofang Ye, Dandan Yang, Sixu Yang, Yong Yin, Renhe Zhang

**Affiliations:** 1Department of Atmospheric and Oceanic Sciences & Institute of Atmospheric Sciences, Fudan University, Shanghai 200437, China; 2Shanghai Typhoon Institute, China Meteorological Administration (CMA), Shanghai 200030, China; 3Shanghai Key Laboratory of Meteorology and Health, Shanghai Meteorological Bureau, Shanghai 200030, China; 4Institute of Occupational Health and Environmental Health, School of Public Health, Lanzhou University, Lanzhou 730000, China; 5Department of Respiratory, School of Medicine, Shanghai Children’s Medical Center, Shanghai Jiao Tong University, Shanghai 200240, China

**Keywords:** PM_2.5_, asthma, pulmonary function, children, panel study

## Abstract

Fine particulate matter (PM_2.5_) has been reported to be an important risk factor for asthma. This study was designed to evaluate the relationship between PM_2.5_ and lung function among children with asthma in Shanghai, China. From 2016 to 2019, a total of 70 Chinese children aged 4 to 14 in Shanghai were recruited for this panel study. The questionnaire was used to collect baseline information, and the lung function covering forced vital capacity (FVC), forced expiratory volume in 1 s (FEV1) and peak expiratory flow (PEF) were carried out for each child more than twice during follow-up. Meanwhile, the simultaneous daily air atmospheric pollutants and meteorological data were collected. The linear mixed effect (LME) model was used to assess the relationship between air pollutants and lung function. A significantly negative association was found between PM_2.5_ and lung function in children with asthma. In the single-pollutant model, the largest effects of PM_2.5_ on lung function were found for lag 0–2, with FVC and FEV1 decreasing by 0.91% [95% confidence interval (CI): −1.75, −0.07] and 1.05% (95% CI: −2.09, 0.00), respectively, for each 10 μg/m^3^ increase in PM_2.5_. In the multi-pollution model (adjusted PM_2.5_ + SO_2_ + O_3_), the maximum effects of PM_2.5_ on FVC and FEV1 also appeared for lag 0–2, with FVC and FEV1 decreasing by 1.57% (95% CI: −2.69, −0.44) and 1.67% (95% CI: −3.05, −0.26), respectively, for each 10 μg/m^3^ increase in PM_2.5_. In the subgroup analysis, boys, preschoolers (<6 years old) and hot seasons (May to September) were more sensitive to changes. Our findings may contribute to a better understanding of the short-term exposure effects of PM_2.5_ on lung function in children with asthma.

## 1. Introduction

As a chronic disorder of the airways characterized by reversible airflow obstruction and airway inflammation, persistent airway hyper-reactivity, and airway remodeling [1], asthma is one of the most common diseases in children [2]. The prevalence of asthma is about 10.5% in 6- to 14-year-old children in 25 countries, and the prevalence rate is still rising [3]. The number of asthma patients worldwide reached 339.44 million in 2016 [4], and low and middle-income countries contributed 96% of global asthma-related deaths and 84% of global disability-adjusted life-years [5]. In China, the average prevalence of asthma in children under 14 years of age increased by 32.7% during 2000 to 2010 [6]. Therefore, it is crucial to study the risk factors of asthma in children to reduce and prevent its risk to children’s health.

In addition to individual-related risk factors such as genetics and tobacco exposure [7], air pollutants, such as nitrogen dioxide (NO_2_), sulfur dioxide (SO_2_), ozone (O_3_), fine particulate matter with aerodynamic diameter ≤ 2.5 μm (PM_2.5_) and particulate matter with aerodynamic diameter ≤ 10 μm (PM_10_) [8,9,10,11,12], were also considered to be related to the occurrence and development of asthma among children [13,14,15]. The most health-harmful pollutant is PM_2.5,_ which could penetrate deep into lung passageways [16] and may trigger an immune response and oxidative stress, which may lead to lung function decrease [17]. However, the results about the effect of PM_2.5_ on asthma have not been consistent, and the biological mechanism of the relationship between the changes of lung function and PM_2.5_ in children with asthma is still unclear [6,18,19,20,21].

A Japanese study showed that each 10 μg/m^3^ increment in PM_2.5_ corresponded to decreased peak expiratory flow (PEF, changes: −2.96, 95% confidence interval: −4.55, −1.37) [22]. Similarly, based on the Columbia Center for Children’s Environmental Health cohort (CCCEH) birth cohort study, PM_2.5_ increase was significantly linked with a decrease in forced expiratory volume in 1 s (FEV1, β: −0.15, 95% confidence interval: −0.29, −0.01) in children with asthma [23]. However, a California study showed that FEV1 in children with asthma decrements were not significantly associated with increasing ambient PM_2.5_ [24]. Additionally, the majority of the research that explored the relationship between lung function and PM_2.5_ based on big data analysis and animal experiments was rarely based on panel study data. Furthermore, China is the largest developing country, with unprecedented socio-economic and urbanization changes, and these changes are associated with a rapidly increasing prevalence of asthma and a significant burden on the health care system [25,26]. Therefore, the effect of PM_2.5_ on lung function in children with asthma needs further study.

Since Shanghai is one of the largest cities in China and has the highest prevalence of childhood asthma in China [27,28], it is necessary to design a panel study to explore the effects of PM_2.5_ on lung function in children with asthma. Therefore, this study explores the relationship between PM_2.5_ and lung function in children with asthma based on a panel study design in Shanghai. The linear mixed effect model was used to estimate these relationships, and subgroup analyses stratified by demographic season were conducted to identify potentially influencing factors. Expectations for providing a scientific basis for the prevention and control of asthma.

## 2. Materials and Methods

### 2.1. Design and Population

From January 2016 to December 2019, we recruited 117 asthmatic children from the Respiratory Department at Shanghai Children’s Medical Center. These participants were included based on the following criteria: (1) no history of chest surgery; (2) asthma cases were extracted from the hospital information systems according to the International Classification of Disease codes, Revision 10 (ICD-10, J45 and J46) or diagnosed by a professional doctor; (3) ages 4–14 years and parent-reported asthma or a history of symptoms including wheezing, cough or dyspnea. Finally, this study included a total of 70 children (39 boys and 31 girls, aged 4–14 years) with asthma for statistical analysis (Figure 1). This study was conducted in accordance with the Declaration of Helsinki. All participants in this study completed questionnaires, lung function measurement and received written informed consent and were approved by the Ethics Committee of Shanghai Children’s Medical Center (SCMCIRB-K2016037).

### 2.2. Lung Function Test and Questionnaire

In this study, the spirometry was performed by experienced operators (medical doctors specialized in pediatric pulmonology) and the specific recommendations for spirometry in the pre-school age were considered [29]: (1) children were instructed how to perform the maneuvers, repeating them at least three times; (2) a training period was considered to familiarize them with the equipment and technician; (3) the operator observed the child closely to ensure there was no leak, and that the maneuver was performed optimally; and (4), to visibly examine volume–time curves and flow–volume curves in real time [30]. The adolescents were all in sitting position and no nasal clips were used in any of the operations.

With standard questionnaires (Appendix A), the trained interviewers collected the data on basic demographic information, parents’ allergy history, allergy history, cigarette exposure and so on. In this study, the frequency of smoking in places where children often move or rest was classified into the following groups: 0 unit/day, <1 unit/day, 1–5 units/day and >5 units/day. The frequency of contact between smokers and children was classified into the following groups: 0 h/day, <1 h/day, 1–4 h/day, 4–8 h/day and >8 h/day. Body mass index (BMI) was calculated as weight (kg) divided by the square of the height (m).

### 2.3. Exposure

The data of daily fine particulate matter with aerodynamic diameter ≤ 2.5 μm (PM_2.5_), particulate matter with aerodynamic diameter ≤ 10 μm (PM_10_), nitrogen dioxide (NO_2_), sulfur dioxide (SO_2_), carbon monoxide (CO), and ozone (O_3_, maximum 8 h average concentration) was obtained from the Shanghai Environmental Monitoring Center. A total of 9 stations of Shanghai Environmental Monitoring Center were used, and the values of each element were taken as the average values of all stations. Daily meteorological data covering temperature (including mean, maximum, and minimum) and relative humidity were provided by the Shanghai Meteorological Bureau (Figure 2).

### 2.4. Statistical Analysis

We set up the questionnaire information database through EpiData (Version 3.0, Odense, Denmark) and used double input for quality control. In the descriptive analysis for baseline characteristics of the asthmatic children and exposure data for environmental factors, categorical variables were described as frequency and percentage, and continuous variables were given as means and standard deviations.

Because there were repeated measures for all participants, the linear mixed effect (LME) model was used to estimate the effects of PM_2.5_ on lung function [30]. This model allows each subject to act as self-control of time and to explain the correlation between repeated measurements collected by each person by including the random effects of the subjects [31]. Due to the abnormal distribution of lung function, the measured values of lung function were transformed logarithmically. We controlled three-day moving average temperature (lag 0–3) and three-day moving average relative humidity (lag 0–3) for potential lagged meteorological confounders. In addition, the time of lung function tests, holidays and day of the week effects were adjusted in the model. The respiratory health indicators were regressed on moving average concentrations of exposure variables from 1 day to 7 days before the measurements to estimate the potential cumulative effects of the exposures. The mixed-effects model is as follows:Log (Y_ij_) = u + b_i_ + β_1_X_1_ + β_2_X_2_ …β_n_X_n_ + ξ_ij_

Log (Y_ij_) represents the logarithmic value of the lung function index for study subject i on measurement j; u represents the fixed intercept; b_i_ is individual-specific random intercept; β_1_ through to β_n_ represent the fixed effect variable coefficients for variables X_1_ through X_n_; and ξ_ij_ represents the error for participant i on measurement j. We reported the results as the estimated percentage change in FVC, FEV1 and PEF with 95% confidence intervals (CI) with each 10 μg/m^3^ increase in air pollutants. The estimated percent changes were calculated as [10^(β × 10)^−1] × 100%; with {95%CI{10^[10 × (β ± 1.96 × SE)]^ − 1} × 100%; where β and SE are the effect estimate and its standard error [32].

We used the single-pollutant model and multi-pollutant model to test the consistency of exposure effects. Pollutants that were highly related to PM_2.5_ (r > 0.7) were excluded in order to reduce the collinearity among pollutants, last SO_2_ and O_3_ were included in the multi-pollutant model (Appendix A). To control for intra-participant variability, the subject was included in the LME model as random variables and adjusted for questionnaire information (atopic dermatitis, allergic rhinitis, food or drug allergies, mother’s or father’s history of allergies, frequency of smoking in places where children often move or rest and frequency of contact between smokers and children), age, sex, and BMI.

We conducted a subgroup analysis by sex (boy vs. girl), age (preschool children were defined as aged less than 6 years old vs. school-aged children defined as aged older than or equal to 6 years) and season (cold season was defined the period of November to March, and the hot season was defined as the period of May to September [20]) to control for potential modifying effects.

The mixed linear effect model was performed with the “lme4” package and “Matrix” package in R software (R Development Core Team; http://R-project.org) [33]. Other statistical analysis was achieved in SPSS 22.0 (IBM SPSS, Chicago, IL, USA), and *p* < 0.05 (two-tailed) was considered statistically significant.

## 3. Results

### 3.1. Descriptive Analysis

Table 1 shows the basic demographic information of the study population. After the filtration of population information, we included 39 boys and 31 girls in this study, and their BMI was 16.55 ± 2.82; 87.14% of children with asthma had allergic rhinitis, 62.86% had atopic dermatitis and 87.14% of parents chose not to smoke in places where their children often had activities or rest. Table 2 shows the descriptive results of the meteorological variables, air pollutants, and lung function indicators. The daily average temperature and relative humidity were 19.17 °C and 74.86%, respectively. The daily average concentrations of PM_2.5_, PM_10_, O_3_, SO_2_, NO_2_ and CO were 34.15 μg/m^3^, 50.69 μg/m^3^, 107.34 μg/m^3^, 9.92 μg/m^3^, 40.38 μg/m^3^ and 663.23 μg/m^3^, respectively. The average of FVC, FEV1 and PEF were 1.63 L, 1.36 L and 3.03 L/s, respectively.

### 3.2. Regression Analysis

Figure 3 shows the changes in lung function of asthmatic children with each 10 μg/m^3^ increase in PM_2.5_ in a single-pollution or multi-pollution model. The changes of lung function in the single-pollution or multi-pollution model were similar, indicating that the model was stable. Delayed effects of PM_2.5_ were significantly associated with FVC and FEV1 on lag 0, lag 0–1 and lag 0–2, both in single-pollution or multi-pollution models, with the largest effect observed on lag 0–2. Per 10 μg/m^3^ increase in PM_2.5_ on lag 0–2 corresponded with a 1.57% (95% CI: −2.69, −0.44) decreased FVC and 1.67% (95% CI: −3.05, −0.26) decreased FEV1 in the multi-pollution model (adjusted PM_2.5_ + SO_2_ + O_3_). However, in the double-pollution model (adjusted PM_2.5_ + O_3_) and the single-pollution model, the delayed effect of PM_2.5_ was significantly related to the effect on PEF on lag 0, with the largest effect of PEF on lag 0–3 in the single-pollution model. For each 10 μg/m^3^ increase in PM_2.5_ on lag 0–3, the PEF decreased 2.12% (95% CI: −4.18, −0.02).

### 3.3. Subgroup Analysis

Figure 4 shows the gender difference in the relationship between lung function and PM_2.5_ in the multi-pollution model (adjusted PM_2.5_ + SO_2_ + O_3_). For boys, the maximum effect of PM_2.5_ on both FVC and FEV1 occurred in lag 0–2, FVC and FEV1 decreased by 3.65% (95% CI: −5.28, −1.98) and 3.12% (95% CI: −5.25, −0.95), respectively, for each 10 μg/m^3^ increase in PM_2.5_. In the girls’ group, the maximum effect of PM_2.5_ on FEV1 occurred on lag 0–4, FEV1 decreased 2.33% (95% CI: −4.38, −0.24) for each 10 μg/m^3^ increase in PM_2.5_. These results were similar to other models (adjusted PM_2.5_, PM_2.5_ + O_3_, PM_2.5_ + SO_2_ in Appendix A).

Figure 5 shows the age difference in the relationship between lung function and PM_2.5_ in the multi-pollution model (adjusted PM_2.5_ + SO_2_ + O_3_). Each 10 μg/m^3^ increase in PM_2.5_ on lag 0–3 was associated with decreased FVC and FEV1 by 3.77% (95% CI: −5.72, −1.77) and 4.25% (95% CI: −6.62, −1.81) in preschool children. As for PEF, each 10 μg/m^3^ increase of PM_2.5_ on lag 0–6 was associated with a maximum effect decrease of 6.89% (95% CI: −10.55, −3.09), but for school-aged children, none of them were statistically significant. The results were similar to other models (adjusted PM_2.5_, PM_2.5_ + O_3_, PM_2.5_ + SO_2_ in Appendix A).

Figure 6 shows the season difference in the relationship between lung function and PM_2.5_ in the multi-pollution model (adjusted PM_2.5_ + SO_2_ + O_3_). No statistically significant relationship between lung function and PM_2.5_ was located in the cold season (November to March). During the hot season (May to September), both FVC and FEV1 had the largest decline when exposed to PM_2.5_ on lag 0–2. For each 10 μg/m^3^ increase of PM_2.5_, FVC decreased by 2.56% (95% CI: −4.74, −0.33, and FEV1 decreased by 3.28% (95% CI: −5.78, −0.72). In the seasonal subgroup analysis, the results were not quite the same for different pollutant models (adjusted PM_2.5_, PM_2.5_ + O_3_, PM_2.5_ + SO_2_ in Appendix A). Among them, the results in the two-pollutants model controlling PM_2.5_ and SO_2_ (Appendix A) were similar to the multi-pollution model (adjusted PM_2.5_ + SO_2_ + O_3_).

## 4. Discussion

In this study, children’s lung function was repeatedly measured, and a linear mixed effect model was used to explore the relationship between air PM_2.5_ and lung function in children with asthma. Mixed-effect models showed that exposure to PM_2.5_ was related to decreases in forced vital capacity (FVC) and forced expiratory volume in 1 s (FEV1), respectively, and there is a cumulative lag effect. In contrast, peak expiratory flow (PEF) change was not associated with PM_2.5_ concentrations. Generally, the change in lung function in boys was more obvious than that in girls. In the subgroup analysis, the changes in lung function in preschool children were more vulnerable to the effects of PM_2.5_ than those in school-age children, and in hot seasons the changes were more obvious than in cold seasons. This study was based on a panel study, and the results may add new knowledge to the effects of PM_2.5_ on lung function in children with asthma aspect.

Consistent with prior studies [34,35,36,37], we also found higher PM_2.5_ was associated with decreased lung function. Xu, et al. [36] assessed the association of lung function with current day PM_2.5_ and found that each 10 μg/m^3^ increment of PM_2.5_ exposure was associated with a 9.83 mL decrease [95% confidence interval (CI): 2.46, 17.19] in FVC and an 8.06 mL decrease (95% CI: 1.06, 15.06) in FEV1; however, associations with PEF were non-statistically significant. In Canada, researchers observed that an interquartile range (IQR) increase (6 μg/m^3^) in the previous 24 h mean concentration of PM_2.5_ was associated with a 0.54% (95% CI: 0.06, 1.02) decrease in bedtime FEV1 (Dales, et al., 2009). Similar results were reported in a Polish study reporting that an IQR increment of PM_2.5_ was related to 2.1% decrease in FVC (95% CI: −2.6, −1.6) and 1.0% decrease in FEV1 (95% CI: −1.4, −0.6) [37]. However, some studies have reported inconsistent results. For example, a systematic review reported that effects of PM_2.5_ on PEF widely spread and more than those for PM_10_ [38]. Epton et al. [35] assessed the association of lung function with air pollution and concluded that no significant correlation between lung function and pollution level could be detected, either in asthmatics or normal students. The inconsistencies among these studies may be attributed by heterogeneity in participants’ characteristics, statistical methodology, other air pollutants, area of study and/or exposure assessment methods. In this study, we used the multi-pollution model to explore the relationship between PM_2.5_ and lung function in children with asthma, but the results were also stable.

These panel study findings imply that PM_2.5_ adversely may affect asthmatic children’s lung function, but the underlying biological mechanism of the effects of PM_2.5_ is still unclear. Some biological mechanisms have been recognized as reasonable ones. Firstly, inhaled PM_2.5_ can trigger oxidative stress and systemic inflammation, which can affect lung function [39,40]. This oxidative stress may also be activating NF-κB and MAPK signaling pathways and promoting the expression of pro-inflammatory factors, which increases nasal mucus and decreases airway barrier function [16,41]. Secondly, a study has confirmed the cytotoxic effect of PM_2.5_ on airway epithelial cells and may activate APCs and T-cells, which can contribute to the exacerbation of respiratory diseases such as asthma [42]. Thirdly, PM_2.5_ exposure can disrupt antioxidant lung function [43], and due to PM_2.5_ having a small diameter which leads to a larger surface area and greater allergen uptake [44], this is more likely to cause a decline in lung function in children with asthma. In this study, we suggested the adversary effects of PM_2.5_ on lung function in children with asthma and the impact of PM_2.5_ may further display potential mechanisms involved in the changes in lung function.

In the subgroup analysis, we found that gender and age may modify the effects of PM_2.5_ on lung function. This study results coincide with the study of Gauderman et al. [45], where a stronger association between PM_2.5_ and lung function change were found in boys. Similarly, the European Study of Cohorts for Air Pollution Effects (ESCAPE) reported that associations between with annual average PM_2.5_ and lung function tended to be somewhat stronger in boys compared with girls [46]. Additionally, Kasamatsu et al. [47] reported that gender had no modification on the effects of PM_2.5_ on PEF. Then, a study in Nanjing suggested that the acute effects of PM_2.5_ on lung function indicators in girls were greater in boys [36]. The effects of modification of gender may be due to gender-linked estrogen [48,49]. This is probably because boys prefer and enjoy being involved in outdoor activities, which causes more frequent exposure to atmospheric pollutants than it does to girls [50]. Sex-related differences in lung function also could be due to the difference in sex hormones and the difference in the airway relative to lung size [51,52].

Additionally, we found a stronger association between lung function and the exposure to PM_2.5_ in the group of preschool children (< 6 years old), which was consistent with other studies [53,54]. This may be explained by the more susceptible status of preschoolers to PM_2.5_ due to their immature immune function and sensitive lung tissue [55,56]. Nevertheless, studies in China showed the effects of PM_2.5_ in 5–14-year-olsd were higher than 0–4-year-olds [57]. Additionally, a study in Hefei [58] found that school-aged children (≥6 years old) were more sensitive than preschoolers (<6 years old) in the relationship between PM_2.5_ exposure and asthma. This may be explained by the different area of air pollution, socioeconomic status and adaptive capacity of children at different locations.

Consistent with the results found in a meta-analysis including 87 studies [21] and a study in Shanghai [44], PM_2.5_ had a greater impact on lung function in children with asthma during the hot season. This result can be explained from three aspects. Firstly, there are higher levels of aerosol molecules compared with the cold seasons, such as fungal spores and pollen, which may trigger more allergy attacks [59,60]. Moreover, a Shanghai study reported that the effects of PM_2.5_ on asthma hospital visits was highest in summer [61]. This may be explained by the high pollution episodes of PM_2.5_ being prone to occur in the eastern Yangtze River Delta of China in summer [62]. Thirdly, children spend more time outdoors and become involved in more physical activities in the warmer season, which increases their per-minute ventilation, and thus they inhale more air pollutants. However, a study conducted in Shenzhen and a time-stratified case-crossover study in the Philadelphia metropolitan region showed that the risk was significantly greater in the cold season than the warm season [63,64]. This is likely to be related to factors such as the chemical composition and levels of ambient particulate matter, the exposure patterns of the local population and local climatic conditions [65,66].

This panel study had many strengths in explaining the association between lung function in children with asthma and short-term exposure to PM_2.5_, but some limitations could not be avoided. Firstly, due to the lack of specific home addresses of participants, the data of air pollutants and meteorological factors from monitoring stations could not represent the specific exposure level for each participant. Secondly, this study only considered the effects of external atmospheric pollutants and meteorological factors, without regard to indoor pollution [13,14,67], which may lead to misclassification and bias. At the same time, one study has suggested that the characteristics of domestic housing conditions and indoor and outdoor pollutants are highly correlated, indicating that the use of outdoor detection data has certain research value [68] and the model also controlled for factors such as the frequency of smoking in places where their children often had activities or rest. What is more, this study did not analyze PM_2.5_ components due to technical limits, which may result in an underestimate of the effect. Fourthly, the sample size was small; however, we performed at least two repeated measurements of lung function and analyzed the data using a mixed-effects model. Last but not least, the analysis of temperature in this study was in a subgroup analysis only, and the interaction between PM_2.5_ and temperature was not considered, which may cause an underestimation of the results.

## 5. Conclusions

In summary, this panel study explored the relationship between atmospheric PM_2.5_ concentrations and lung function in children with asthma. The single- and multi-pollutant models all suggested that the increase in PM_2.5_ concentration was associated with the decline of lung function and short-term exposure to PM_2.5_ may affect the health of lung function in children with asthma, particularly in boys, preschoolers (<6 years old) and in the hot seasons (May to September). The results suggest that more attention should be paid to the effect of PM_2.5_ on the lung health of asthmatic children. In addition, the effects of PM_2.5_ on lung function showed significant lag effects. The results suggested that the importance of protective measures after air pollution events, such as wearing masks, reducing non-essential going out and so on.

## Figures and Tables

**Figure 1 ijerph-19-11385-f001:**
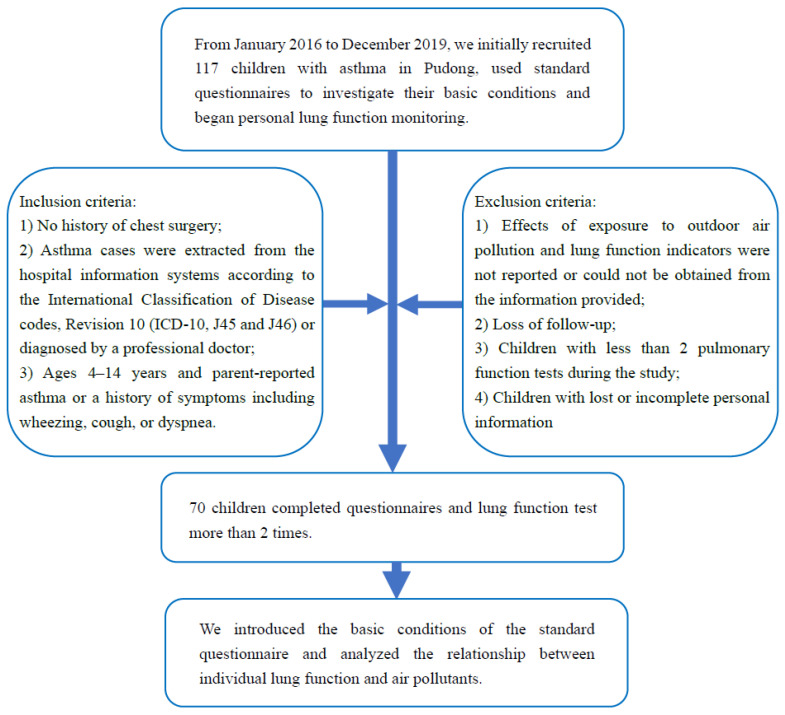
The selection of the participants in this study.

**Figure 2 ijerph-19-11385-f002:**
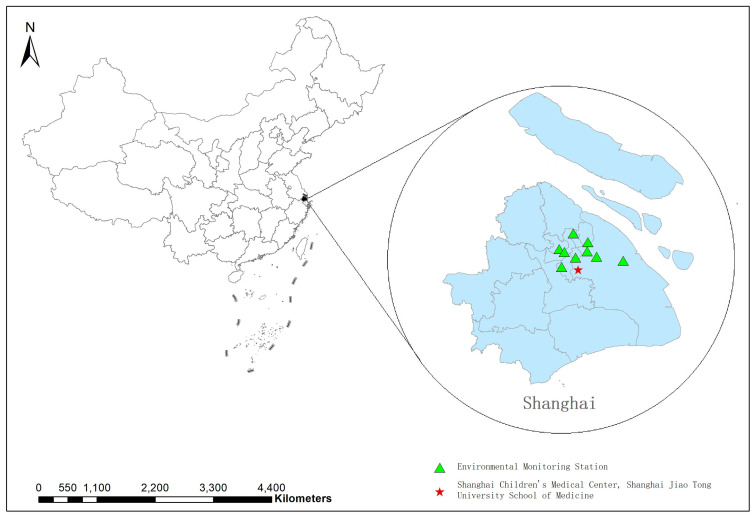
Location of environmental monitoring station and Shanghai Children’s Medical Center, Shanghai Jiao Tong University School of Medicine in Shanghai.

**Figure 3 ijerph-19-11385-f003:**
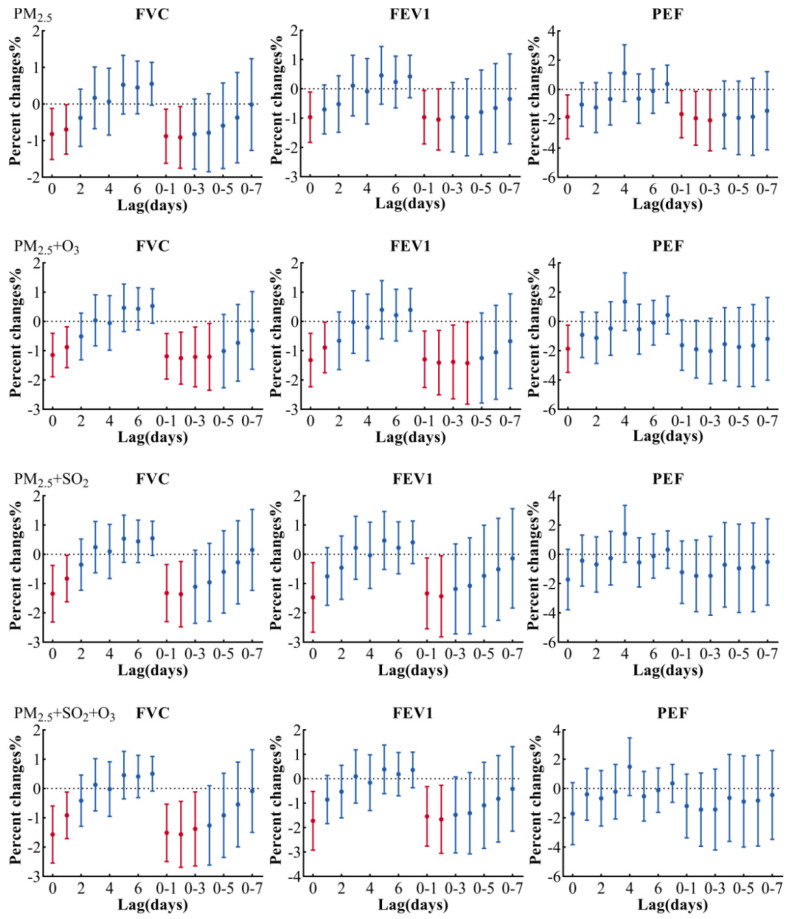
Association between lung function and air pollutants in single- or multi-pollution models. Estimates are adjusted for questionnaire information, relative humidity, temperature, age, BMI, sex, holiday, day of week and times of the measurement. PM_2.5_, fine particulate matter with aerodynamic diameter ≤ 2.5 μm; O_3_, ozone; SO_2_, sulfur dioxide; FVC, forced vital capacity; FEV1, forced expiratory volume in 1 s; PEF, peak expiratory flow.

**Figure 4 ijerph-19-11385-f004:**
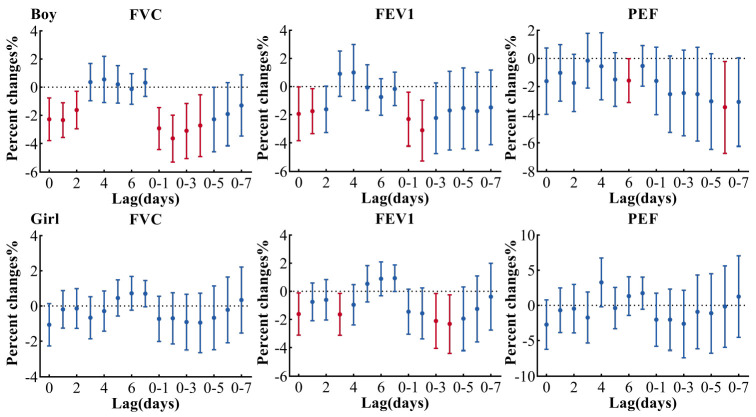
Association between lung function and PM_2.5_ in gender-stratified multi-pollution model (adjusted PM_2.5_ + SO_2_ + O_3_). Estimates are adjusted for questionnaire information, relative humidity, temperature, age, BMI, holiday, day of week and times of the measurement. FVC, forced vital capacity; FEV1, forced expiratory volume in 1 s; PEF, peak expiratory flow.

**Figure 5 ijerph-19-11385-f005:**
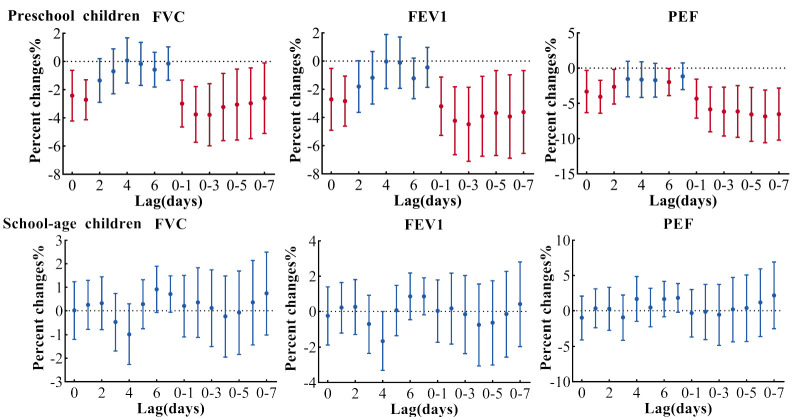
Association between lung function and PM_2.5_ in age-stratified multi-pollution model (adjusted PM_2.5_ + SO_2_ + O_3_). Preschool children: age < 6 years old; school-age children: age ≥ 6 years old. Estimates are adjusted for questionnaire information, relative humidity, temperature, BMI, sex, holiday, day of week and times of the measurement. FVC, forced vital capacity; FEV1, forced expiratory volume in 1 s; PEF, peak expiratory flow.

**Figure 6 ijerph-19-11385-f006:**
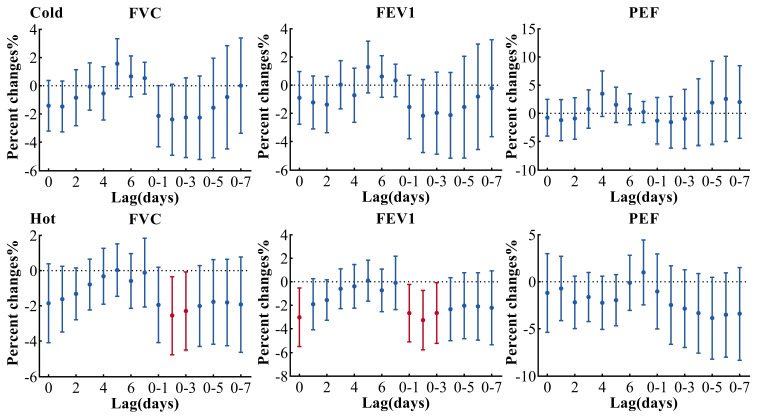
Association between lung function and PM_2.5_ in a season-stratified multi-pollution model (adjusted PM_2.5_ + SO_2_ + O_3_). Cold season refers to the period of November to March, and hot season means May to September [20]. Estimates are adjusted for questionnaire information, relative humidity, temperature, age, BMI, sex, holiday, day of week and times of the measurement. FVC, forced vital capacity; FEV1, forced expiratory volume in 1 s; PEF, peak expiratory flow.

**Table 1 ijerph-19-11385-t001:** Description of basic demographic information (*n* = 70).

	*n*	%
Sex		
boy	39	55.71
girl	31	44.29
Age		
<6	40	57.14
≥6	30	42.86
BMI，Mean ± SD	16.55 ± 2.82	
Atopic dermatitis		
Yes	44	62.86
No	26	37.14
Allergic rhinitis		
Yes	61	87.14
No	9	12.86
Food or drug allergies		
Yes	20	28.57
No	50	71.43
Mother’s history of allergies		
Yes	52	74.29
No	18	25.71
Father’s allergy history		
Yes	52	74.29
No	18	25.71
Frequency of smoking in places where their children often had activities or rest		
0 units/day	61	87.14
<1 unit/day	2	2.86
1~5 units/day	6	8.57
>5 units/day	1	1.43
Frequency of contact between smokers and children		
0 h/day	43	61.43
<1 h/day	9	12.86
1–4 h/day	11	15.71
4–8 h/day	3	4.29
>8 h/day	4	5.71

**Table 2 ijerph-19-11385-t002:** Description of basic meteorological pollutants and lung function indicators.

	Mean	SD	Min	Max	IQR
Daily temperature (℃)	19.17	9.03	−0.10	32.20	16.40
Relative humidity (%)	74.86	11.73	37.00	97.50	18.30
PM_2.5_ (μg/m^3^)	34.15	25.13	9.00	173.00	28.00
PM_10_ (μg/m^3^)	50.69	26.46	13.00	166.00	34.00
O_3_ (μg/m^3^)	107.34	48.04	32.00	251.00	67.00
SO_2_ (μg/m^3^)	9.92	4.30	5.00	40.00	5.00
NO_2_ (μg/m^3^)	40.38	19.84	13.00	118.00	26.00
CO (μg/m^3^)	663.23	236.13	400.00	1800.00	200.00
FVC (L)	1.63	0.52	0.57	3.85	0.62
FEV1 (L)	1.36	0.42	0.54	3.45	0.51
PEF (L/s)	3.03	1.61	1.05	21.80	1.43

SD, standard deviation; IQR, interquartile range; PM_2.5_, fine particulate matter with aerodynamic diameter ≤ 2.5 μm; PM_10_, particulate matter with aerodynamic diameter ≤ 10 μm; O_3_, ozone; SO_2_, sulfur dioxide; NO_2_, nitrogen dioxide; CO, carbon monoxide; FVC, forced vital capacity; FEV1, forced expiratory volume in 1 s; PEF, peak expiratory flow.

## Data Availability

The meteorological datasets and air pollutant datasets used and/or analyzed are available from the open-access websites. All data on children with asthma were obtained from the Shanghai Children’s Medical Center.

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
