# Peer review of "The Effects of Short-Term PM2.5 Exposure on Pulmonary Function among Children with Asthma—A Panel Study in Shanghai, China"

_ijerph, 2022, doi:10.3390/ijerph191811385_

Round 1

Reviewer 1 Report

This study determined the effects of short-term PM2.5 exposure on pulmonary function among children with asthma in Shanghai, China. The paper is well written and the study is conducted comprehensively. The paper contains interesting data which appear to be valid. However, many similar studies have been performed, the highlight or innovative of the study should be addressed. In response to this manuscript, I have the following comments:

Line 98 to 99: “1) children were instructed how to do the manoeuvres, repeating them at least three times”.

Comment: What criteria do you refer to ensure each manoeuvre as qualified? Could you introduce briefly how do you utilize the three manoeuvres to do later analysis, for example, do you choose the mean value of three test? Or you just extract the optimal manoeuvre from three tests? Because lung function value is used as outcome in the later analysis, it is important to elucidate clearly about how you make use of these three test values.

Line 157 to 158: “teenager was defined as aged older than or equal to 6 years old”.

Comment: Actually, I know you want to divide all children into two groups with balanced sample size, but I think defining teenager with age of more than 6 years old is inappropriate because almost no literature used this criterion to define teenager. If you used this definition here, it is better to cite the relevant reference. Or you could rename the subgroups of children with preschool children and school-age children.

Line 194 to 196: “the delayed effect of PM2.5 was significantly related to the effect of PEF on lag 0, with the largest effect of PEF on lag 0-3 in the single-pollution model.”

Comment: I think there are typos here. Do you want to express that PM2.5 concentration on lag0 was significantly associated with PEF, and the largest effect of PM2.5 on PEF was on lag0-3? Therefore, “the delayed effect of PM2.5 was significantly related to the effect of PEF on lag 0”, in this sentence, there are two “effect of” that make confusions. I think the correct expression maybe “the delayed effect of PM2.5 was significantly related to PEF on lag 0”. And “with the largest effect of PEF on lag 0-3 in the single-pollution model” may be “with the largest effect on PEF on lag 0-3 in the single-pollution model”. The noun after “effect of” means something that causes an effect like PM2.5, and the noun after “on” means a result caused by this effect like PEF.

Line 196 to 197: “For each 10 μg/m3 increase in PM2.5 on lag 0-3, the PEF decreased 2.12% (95% CI: -4.18, -0.02).”

Comment: You stated that in one-pollutant model, the effect of PM2.5 on PEF was significant on lag0-3. But in the Figure 2, the interval plot of lag0-3 PM2.5 on PEF is blue but not red that indicates non-significance.

Line 207 to 209: “For boys, the maximum effect of PM2.5 on both FVC and FEV1 occurred in on lag 0-2, FVC and FEV1 decreased by 3.65% (95% CI: -5.28, -1.98) and 3.12% (95% CI: -5.25, -0.95). respectively for each 10 μg/m3 increase of PM2.5.

Comment: There is a redundant full stop before “respectively”.

Line 279 to 282: “But unlike the study in Hefei (Y. Zhang et al., 2019), PM2.5 has a stronger effect on school-age children (6 years old) than preschoolers (<6 years old). This may be explained by the higher level of air pollutants like NO2, which may affect the lung function differently.”

Comment: Could you specify more clearly which study had higher NO2 level? And why PM2.5 together with different level of NO2 could induce different effects on lung function in different age groups? Do you mean that in Hefei’s study, PM2.5 combined with higher level of NO2 affected more on school-age children? But in your study, PM2.5 combined with relatively lower levels of NO2 affected more on preschoolers? Could you find some references to support your opinion?

Reviewer 2 Report

The reviewer has gone through the whole manuscript and recommends some of the major revisions required.

The comments of the reviewer are given below:

Ø  Please avoid the personal manner of addressing (we, our) and replace those words with impersonal (they, these, those, the present study/paper/research, etc). Please also revise the entire manuscript in this regard (L88, L307). The paper will sound more professional.

Ø   The Introduction portion should be rewritten in a systematic way so that its content would be in line with the specific objective of the study. Some focus should be given to explaining the scientific importance and linkage of lung function measurement (FVC, FEV1, and PEF) and children's lung health.

Ø  A study area map can enhance the clarity of the location and its surroundings.

Ø  L32: “In the subgroup analysis… doesn’t gives proper meaning.

Ø  L42: The quoted reference seems old (2007) which leads to less data/value, Use recent reference.

Ø  L103: Use sitting position instead of sat position.

Ø  L106-110: Does the classification have any linked/old references?

Ø  L117: Does High temp, Low temp shows the Max-Min temp?

Ø  Which type of questions was there in the questionnaires, it’s good to provide questionnaires in the supplementary information.

Ø  L206: “Generally, the change of lung function in boys was more obvious than that in girls” Explain how you conclude that. Support with proper reference.

Ø  Line 251-253: Since the associations probably involved multiple influencing factors and conditions, thus a simple comparison is somewhat not meaningful.

Ø  L305: “Finally all health data…” this is not a scientific meaning/finding to discuss.

Ø  L311: “Secondly, this study only considered the effects …. without regard to indoor pollution”. This Limitation is very wide and much important in the case of children, especially 4-10 years who spend most of their time Indoor.

Ø  The Conclusion should be more elaborative.

Reviewer 3 Report

It is necessary to briefly describe the methodology used in this study in the last paragraph of the introduction.

Novelty or originality should be emphasized in Introduction. 

2.1
Describe how the ethical part could be addressed in this study.

3.1

It is unclear if the main cause of asthma is air pollutants or fine dust.

Discussion with investigated data should match with the title focusing on PM2.5

Discussion

It would be better to observe the correlationship between Asthma and PM2.5 based on the specific characteristics of the Shanghai area.

 (Y. Liu et al., 2020; Z.... check the reference description ?

315; I don't think it's necessary to mention the temperature effect in this study.

Throughout the manuscript, the discussion for the study seems insufficient.

Round 2

Reviewer 3 Report

Data graph should be clearer.